# The impact of alloying on defect-free nanoparticles exhibiting softer but tougher behavior

Anuj Bisht [1], Raj Kiran Koju [2], Yuanshen Qi[1], James Hickman[3], Yuri Mishin [2✉] & Eugen Rabkin [1✉]

The classic paradigm of physical metallurgy is that the addition of alloying elements to metals increases their strength. It is less known if the solution-hardening can occur in nano-scale objects, and it is totally unknown how alloying can impact the strength of defect-free faceted nanoparticles. Purely metallic defect-free nanoparticles exhibit an ultra-high strength approaching the theoretical limit. Tested in compression, they deform elastically until the nucleation of the first dislocation, after which they collapse into a pancake shape. Here, we show by experiments and atomistic simulations that the alloying of Ni nanoparticles with Co reduces their ultimate strength. This counter-intuitive solution-softening effect is explained by solute-induced local spatial variations of the resolved shear stress, causing premature dislocation nucleation. The subsequent particle deformation requires more work, making it tougher. The emerging compromise between strength and toughness makes alloy nano-particles promising candidates for applications.

[1] Department of Materials Science and Engineering, Technion-Israel Institute of Technology, Haifa, Israel. [2] Department of Physics and Astronomy, MSN 3F3, George Mason University, Fairfax, VA, USA. [3] Materials Science and Engineering Division, National Institute of Standards and Technology, Gaithersburg, MD, USA. ✉email: ymishin@gmu.edu; erabkin@technion.ac.il

The role of the theoretical strength of crystalline materials is similar to the role of the speed of light in physics: it cannot be reached but sets the physical limits of what is possible. Achieving the theoretical strength of a metallic material is the holy grail of physical metallurgy. The theoretical strength of metals generally falls in the range of $G/30 – G/8$, where $G$ is the elastic shear modulus of the metal[1]. In reality, metals and alloys yield plastically at much lower stresses. Their plastic deformation is governed by the motion of both new and existing dislocations and the activation of new internal dislocation sources. Several strategies have been explored to increase the materials' strength, such as alloying[2], precipitation strengthening[3], grain size reduction[2], grain boundary engineering[4], and microstructure tailoring[5,6]. Although these strategies do help improve the strength, it still remains far below the theoretical limit.

Sixty years ago, Brenner[7] discovered that the strength of metallic whiskers increases with decreasing diameter from several micrometers downscale, eventually reaching the GPa level. The increase in strength is primarily attributed to a change in the plastic deformation mechanism from dislocation motion and multiplication in bulk materials to dislocation nucleation in defect-free samples of sub-micrometer dimensions. Metallic micro- and nanoparticles obtained by solid-state dewetting are examples of defect-free metallic objects with strength approaching the theoretical limit[8–10]. They exhibit significantly higher compression strength than defect-free metallic whiskers tested in tension[11,12]. In the latter case, the large surface area catalyzes the heterogeneous nucleation of dislocation half-loops at much smaller stresses than can be supported by nanoparticles tested in compression[13,14].

Recently, a record-high strength of 34 GPa was reported for defect-free face-centered cubic (FCC) Ni nanoparticles produced by solid-state dewetting[9]. The particles had a faceted shape with relatively rounded corners and edges and the (111) top facet aligned parallel to the substrate. During the compression, the particles showed elastic behavior up to a strain of about 0.2, followed by a sudden collapse into a pancake shape. The rounded edges and corners reduced the stress concentration and delayed the collapse until stresses approaching the theoretical limit. An even higher compressive strength of 46 GPa (the highest strength ever reported for metallic materials) was recently achieved in body-centered cubic Mo microparticles produced by two-stage solid-state dewetting[10]. The deformation mode and the size exponent were similar to those in the FCC Ni nanoparticles, suggesting that the absence of defects and the particle shape are more important for the strength than the crystalline structure. Molecular dynamics (MD) simulations[9,15] have helped understand the dislocation mechanisms responsible for the strength of defect-free nanoparticles.

Previous studies of defect-free nanoparticles were focused on pure metals[8–10,12]. Meanwhile, most technological applications utilize alloys rather than pure metals. Alloying is the standard way of improving materials' strength in conventional metallurgy. In bulk alloys, the solute atoms act as pinning centers, hindering the dislocation motion and increasing the strength. However, the effect of alloying on the nucleation-controlled deformation of nanoparticles remains largely unknown. The huge elastic energy accumulated during the elastic stage of deformation can propel the newly nucleated dislocations through the particle at high speed. It is not clear a priori if the traditional solid-solution hardening mechanisms can operate under such extreme conditions. Furthermore, the solutes can affect the particle strength by impacting the dislocation nucleation process. For example, Zou et al.[16] suggested that the coherency strains induced by misfitting solute atoms segregating to the surface in alloy thin films can make the dislocation nucleation easier than in pure metals. This

"solute softening" effect was observed in recent MD simulations of Fe-Ni nanowires[17] and was attributed to local pressure fluctuations near the surface. Similarly, MD simulations of an isolated spherical pore in Mg have shown a decrease in the spall strength with the addition of Al atoms[18]. This spall softening effect was explained by atomic misfit stresses promoting the dislocation nucleation at the pore surface. However, the solute effect on the strength of defect-free metallic nanoparticles has never been studied experimentally or by simulation.

Here, we investigate the effect of alloying on the strength of Ni-Co nanoparticles. Co exhibits an unlimited solubility in FCC Ni at high temperatures[19]. This allows us to focus this research on the solid-solution effect uninfluenced by the precipitation hardening and other mechanisms of alloy strengthening. Based on the knowledge of bulk Ni-Co alloys and Ni-Co wires, one would expect that the addition of Co should increase the strength of the nanoparticles. Indeed, Ni-Co alloys perfectly follow the classic pattern of solid-solution hardening both in the bulk form and as wires. The yield stress of well-annealed wires of 1 mm in diameter increases from 47.7 MPa in pure Ni to 68.3 MPa in the Ni-0.3Co weight fraction alloy, with the flow stress increasing from 176 to 262 MPa[20]. MD simulations predict that the Ni-Co nanowires' strength increases to about 10 GPa upon the addition of 0.05 to 0.10 mole fractions of Co[21]. Contrary to this expectation, we find here that the strength of defect-free Ni-Co nanoparticles actually decreases with the addition of Co, resulting in the counter-intuitive solute-softening effect. Our MD simulations reveal that the softening is caused by increased statistical variations of the local resolved shear stress in near-surface regions of the particles, triggering early nucleation of the first dislocation. At the same time, we show that the particles' toughness increases with the addition of Co, creating a combination of strength and toughness that can be attractive for technological applications.

## Results

**Experimental results**. We start by presenting the experimental results of this work. The nanoparticles were fabricated by solid-state dewetting of a 30 nm thick Ni-Co bilayer film from a single-crystalline sapphire substrate (Fig. 1a). Two films with the target compositions of Ni-0.3Co and Ni-0.5Co (in mole fractions), referred to hereafter as Ni-0.3Co and Ni-0.5Co, respectively, were produced. A pure Ni film of the same thickness was also deposited for reference. The nanoparticles formed as a result of dewetting the film from the substrate at the temperature of 1150 °C (see Fig. 1b for the heat treatment protocol and the "Methods" section for details of the fabrication process).

The HR-SEM micrographs in Fig. 2a–c show that many of the particles obtained have faceted near-equilibrium shapes[8,9], although some have not reached equilibrium and remain elongated parallel to the substrate[22,23]. All particles exhibit the same set of facets with rounded edges and corners. The shape and morphology of the particles are similar for all three compositions. To confirm the chemical homogeneity of the particles, we examined TEM lamellae cut through the particle centers and aligned with the three-fold symmetry plane. The bright-field (BF) TEM micrographs, SAEDs, and the EDS elemental maps of the cross-sections are shown in Fig. 2d, e. The BF micrographs reveal the bottom facets not visible in the top-view SEM micrographs. These facets indicate that the particle center is located above the substrate in good agreement with the previous findings[9,24]. SAED patterns obtained for the [110] zone axis exhibit sharp spots indicating the crystalline nature of the particles. The particle facets identified with SAED are mainly parallel to {200} and {111} crystallographic planes marked in Fig. 2d, e. Occasionally, {112} facets could also be found in the rounded regions between other

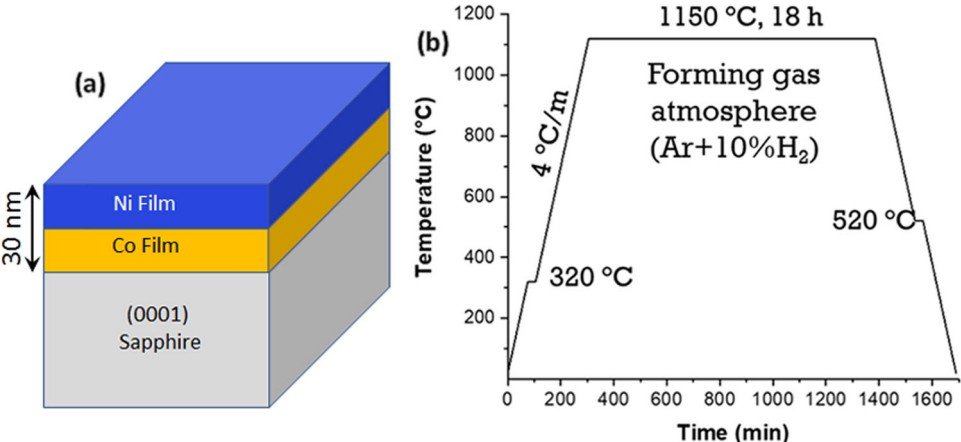

**Fig. 1 Fabrication of defect-free Ni-Co nanoparticles by solid-state dewetting. a** Schematic of the initial Ni-Co bilayer thin film on (0001)-oriented sapphire substrate. **b** Heat treatment schedule employed for the dewetting of the film.

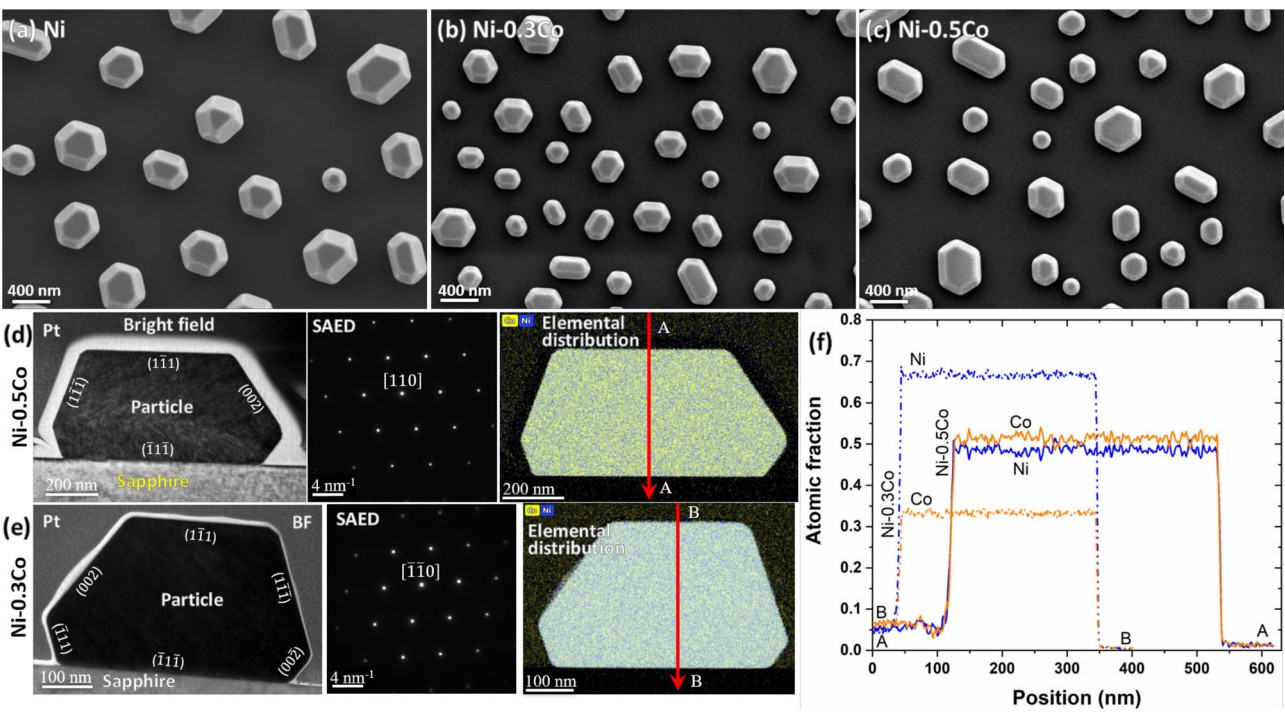

**Fig. 2 Characterization of Ni and Ni-Co nanoparticles produced by solid-state dewetting.** SEM micrographs of faceted single-crystalline (**a**) Ni, (**b**) Ni-0.3Co, and (**c**) Ni-0.5Co nanoparticles obtained by solid-state dewetting. **d, e** Bright-field TEM micrographs, selected-area electron diffraction (SAED) patterns, and TEM EDS elemental distribution maps (Blue: Ni; Yellow: Co) for (**d**) Ni-0.3Co and (**e**) Ni-0.5Co nanoparticles. **f** Concentration line profiles across the particles along the lines AA and BB marked in (**d**) and (**e**).

low-index facets, but they are much smaller. The particles' top surface corresponds to the {111} crystallographic plane and is parallel to the (0001) plane of the substrate. The orientation relationship between the particle and the substrate is further evident from the HR-STEM micrograph of the particle-substrate interface in Supplementary Fig. 1. The elemental maps (Fig. 2d, e) and composition profiles (Fig. 2f) confirm the homogeneous distribution of Ni and Co atoms throughout the particles and the absence of any segregation at the particle surface or the particle-substrate interface. The average Co concentrations in the Ni-0.3Co (0.333 mole fraction) and Ni-0.5Co (0.514 mole fraction) particles are close to the nominal compositions of the initial Ni-Co bilayers. While most particles are single-crystalline, some Ni-0.5Co particles contain a twin boundary parallel to the substrate.

Compression tests were performed in-situ within SEM. Most particles exhibited purely elastic response up to the engineering strain of several percent, followed by an abrupt strain burst previously observed in metallic particles[8–10] (Supplementary Fig. 2). Some particles exhibited a "staircase" yielding, probably caused by trapped internal defects[25]. Such cases were excluded from further analyses. The engineering stress-strain curves obtained for different particle sizes are summarized in Fig. 3a–c. The engineering stress was obtained by dividing the load by the top facet area in contact with the indenter. The engineering strain was calculated by dividing the indenter's displacement by the initial height of the particle from AFM measurements. This engineering strain overestimates the true particle strain as it includes the substrate deformation and machine compliance. The

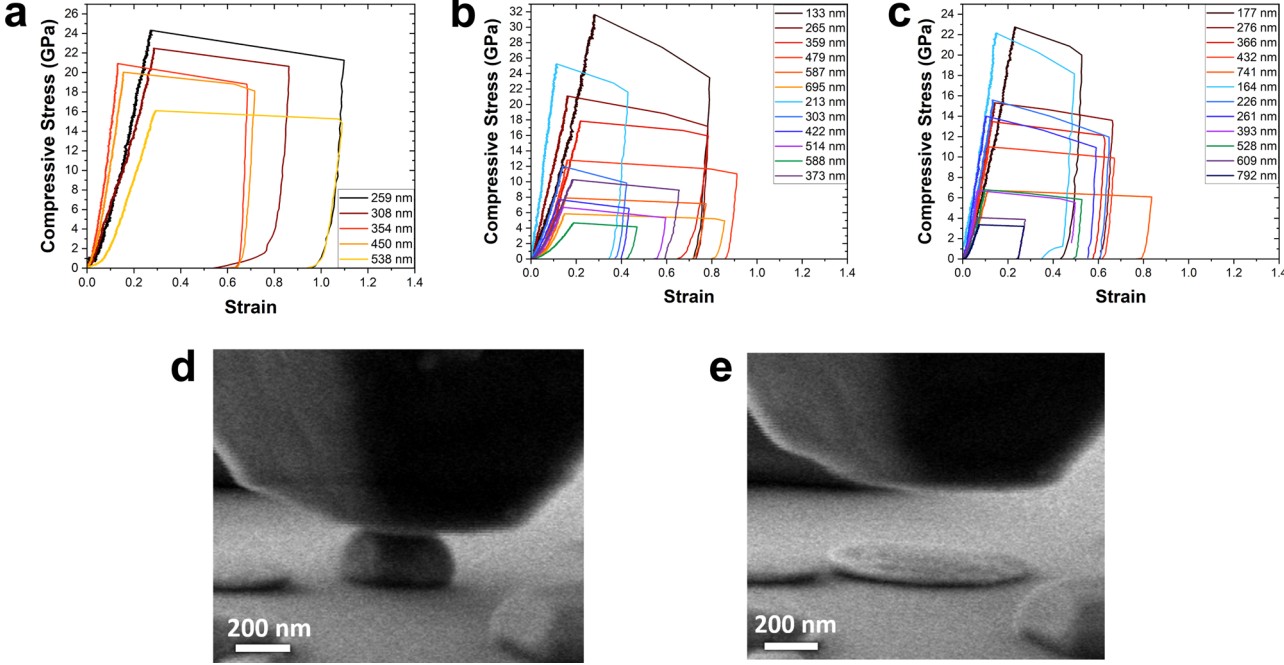

**Fig. 3 Mechanical testing of Ni and Ni-Co nanoparticles. a–c** Engineering compressive stress-strain curves measured in this work for selected particles of (**a**) Ni, (**b**) Ni-0.3Co, and (**c**) Ni-0.5Co. **d,e** SEM image of a Ni-0.3Co particle (**a**) during a compression test and (**b**) after collapse into a disk causing a stain burst.

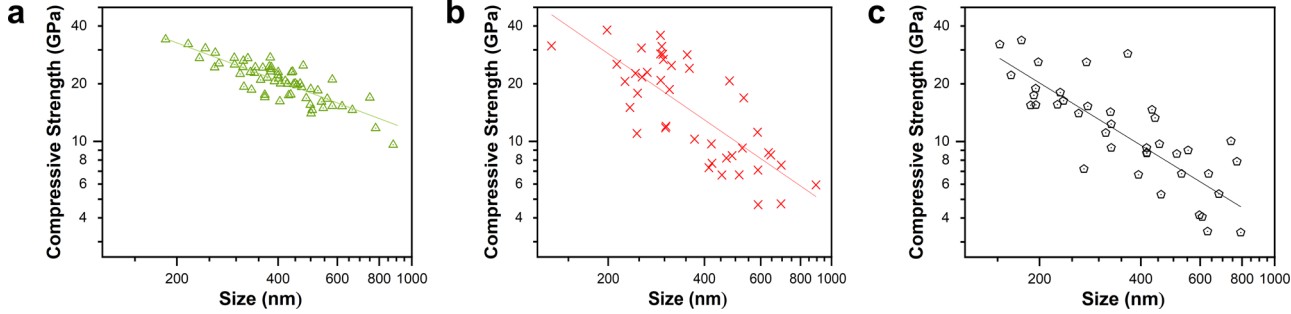

**Fig. 4 Effect of alloying on the mechanical strength of nanoparticles.** Experimental compressive strength as a function of particle size for (**a**) Ni, (**b**) Ni-0.3Co, and (**c**) Ni-0.5Co. The straight lines show linear fits.

effective particle size is defined as the square root of the total projected area of the particle determined from SEM micrographs. The strain bursts in Fig. 3a–c are caused by the onset of plastic deformation and the accompanying drop in the material's resistance to the load. Once the resistance is lost, the indenter drives forward and smashes the particle into a disk (pancake) (Fig. 3e).

We define the compressive strength $\sigma$ of a particle as the engineering stress at the onset of plasticity. The strength dependence on the particle size $d$ is displayed in Fig. 4, where the new data for pure Ni is combined with the previous results[9]. Despite the large scatter and significant overlap of the three datasets, the softening trend upon the addition of Co is apparent and is especially pronounced for larger particles. The size exponents $n$ obtained from $\sigma \propto d^{-n}$ fits of the datasets are 0.70 ± 0.05, 1.15 ± 0.15, and 1.08 ± 0.12 for pure Ni, Ni-0.3Co, and Ni-0.5Co, respectively (the error bars represent one standard deviation). Although the confidence intervals of the alloy dataset overlap, the average values of $n$ suggest a non-monotonic behavior with the Ni-0.3Co composition exhibiting a stronger size dependence than the Ni-0.5Co composition. The atomistic simulations discussed below reveal a similar non-monotonic

behavior of the softening effect and help understand the underlying mechanisms.

**Results of atomistic simulations.** We next present the results of atomistic simulations that were designed to provide complementary information on the microscopic mechanisms of the nucleation-controlled deformation. Atomic interactions in the Ni-Co system were described by a well-tested angular-dependent many-body interatomic potential[26]. The computer-generated nanoparticles had faceted shapes with rounded corners and edges mimicking the experimental shapes. The chemical composition of the particles varied from pure Ni to Ni-60Co. The simulated compression tests were performed by clamping the particle between two semi-soft walls, one of which was fixed and the other moved into the particle with a constant speed. The top and bottom (111) facets were parallel to the walls. The engineering strain and stress were defined as in the experiment. Further simulation details can be found in the "Methods" section.

As in the experiments, the particles deformed elastically up to a few percent of strain followed by an abrupt drop of the stress signifying the onset of plastic deformation (Fig. 5a). The size

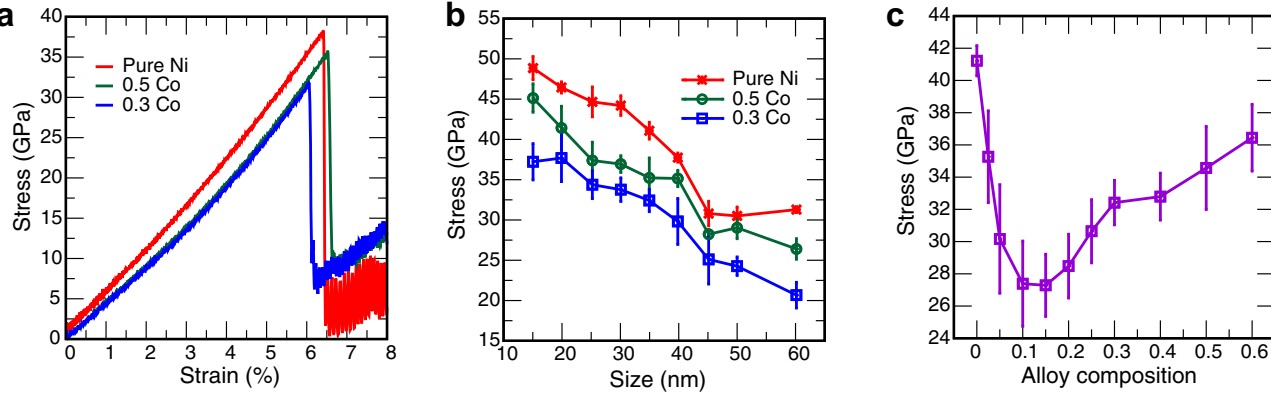

**Fig. 5 Mechanical properties of nanoparticles. a** Typical engineering stress-strain curves for a 40 nm particle. **b** Strength as a function of particle size. **c** Strength as a function of chemical composition for 35 nm particles. The error bars represent one standard deviation.

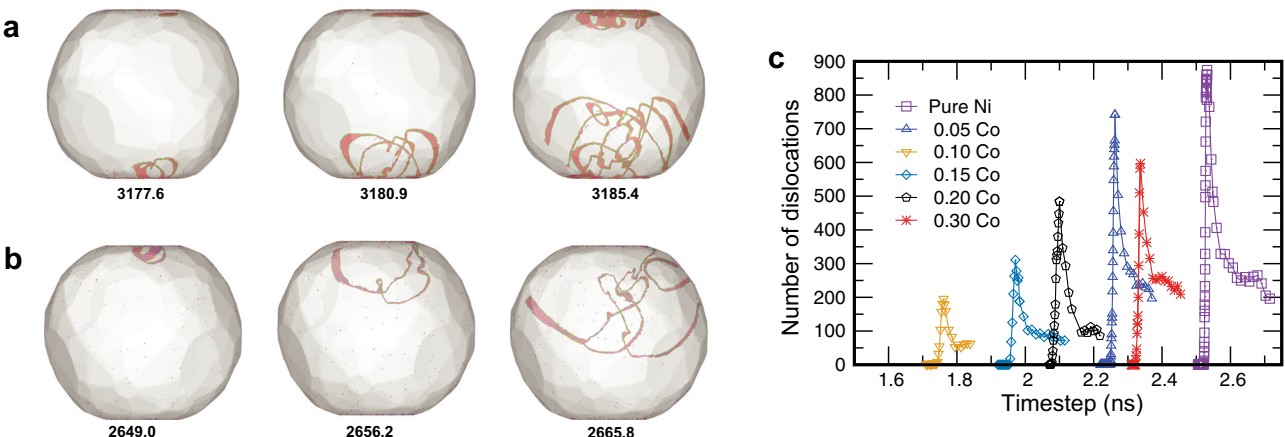

**Fig. 6 Dislocation processes at early stages of plastic deformation.** Dislocation nucleation and multiplication in 60 nm nanoparticles of (**a**) Ni and (**b**) Ni-0.5Co. The numbers indicate the time (in picoseconds) from the start of the simulation. The dislocations are visualized by the DXA algorithm with the red atoms representing stacking faults. Non-dislocated atoms are invisible. **c** Number of dislocation segments in 35 nm nanoparticles for several chemical compositions.

dependence of the particle strength, defined as the peak stress, is shown in Fig. 5b. The results follow the smaller is stronger trend observed experimentally (cf. Fig. 4) but extend this trend to smaller sizes not covered by the experiment. Note that Co reduces the particle strength in a non-monotonic manner: the Ni-0.3Co particles have a smaller strength than both pure Ni and the Ni-0.5Co particles. This non-monotonic behavior of the strength correlates with the experiment and is presented in more detail in Fig. 5c. The strength reaches a minimum at about (0.10–0.15) Co, followed by an increase at higher concentrations.

On the dislocation level, plastic deformation always started with the nucleation of a partial and then a full dislocation half-loop at the top or bottom facet, usually close to one of its edges. Occasionally, two dislocations nucleated almost simultaneously. The subsequent events depended on the chemical composition. In pure Ni particles (Fig. 6a), as the first dislocation half-loop grew, more dislocations nucleated on the same facet, creating a dense dislocation crowd propagating deeper into the particle. The dislocation segments had the form of smooth arcs with the usual dissociation into partials. In some cases, a similar dislocation crowd emerged at the opposite facet and also propagated into the particle. The dislocations quickly spread across the particle and formed a tangle with a high dislocation density. This point marked the end of the first stress drop observed in the simulations and the strain burst in the experiments. The dislocations then annihilated with each other and/or at the side

surfaces and their density decreased, waiting for the next (usually smaller) deformation burst. In the alloy particles (Fig. 6b), the nucleation of the first dislocation (occasionally, two dislocations) happened earlier in time. Only this first dislocation usually propagated into the particle without any other nucleation events. The dislocation line was broken into multiple segments separated by pinning points. It soon developed a highly entangled shape and left debris in the form of tiny loops and other defects in its wake. New dislocations only nucleated when this tangle reached the opposite facet and/or spread over the side surfaces. The maximum dislocation density reached was smaller than in pure Ni particles (Fig. 6c), which correlates with the smaller strain bursts in the experimental alloy particles (cf. Fig. 3a–c).

Having established the solute softening effect by both experiments and simulations, we next turn the attention to the deformation after the stress drop. This stage could not be examined in the experiments due to the inertia of the indenter that immediately smashed the particle. However, the simulations reveal that the subsequent compression of the particles is accompanied by a series of smaller stress peaks similar to a saw tooth (Fig. 7a). The bursts of plastic deformation are caused by dislocation avalanches followed by periods of elastic energy accumulation under a relatively low dislocation density. On average, the stress displays a clear increasing trend with the strain, an effect known as strain hardening. The area under the stress-strain curve from zero to a particular strain is the work of

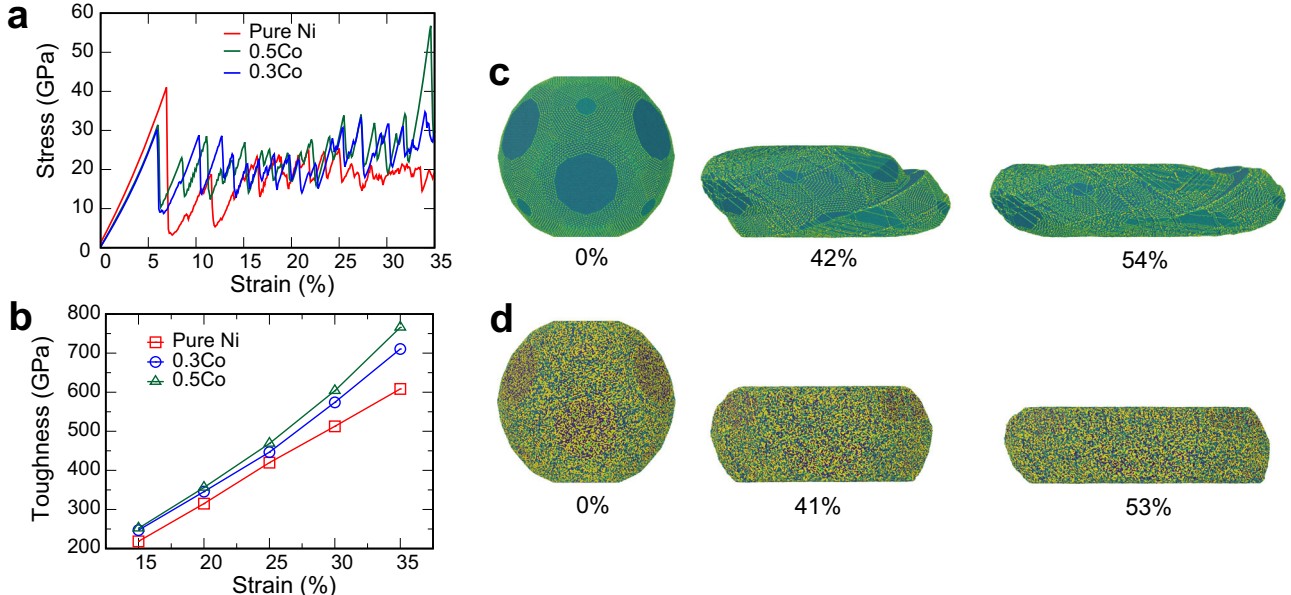

**Fig. 7 Alloying increases the nanoparticle toughness. a** Example of stress-strain curves for 35 nm nanoparticles deformed up to the strain of 350.5Co. **b** Toughness of 35 nm nanoparticles as a function of strain. **c–d** Deformed shapes of the **c** Ni and **d** Ni-0.5Co 35 nm nanoparticles. The atoms are colored according to potential energy, with brighter colors representing larger energy. The strain is indicated under the images.

deformation commonly referred to as toughness. Fig. 7b and Supplementary Fig. 3 show that the toughness increases monotonically with the Co concentration, especially as more strain is accumulated. In other words, while Co reduces the particle strength, it also increases its toughness. The plastic deformation starts earlier but requires more work to continue. The alloying also changes the character of plastic deformation. In pure Ni particles, the plastic flow is unstable and easily localizes in shear bands causing irregular particle shapes (Fig. 7c). By contrast, the deformation of the alloy particles is more homogeneous and preserves the faceted shape down to strong compressions (Fig. 7d).

## Discussion

The experiments and simulations reported here demonstrate that alloying of defect-free Ni nanoparticles with Co reduces their strength. This counter-intuitive solid-solution softening effect contradicts the established behavior of bulk alloys, which are dominated by the solid-solution hardening[20]. At the same time, the alloying makes the particles tougher, increases the strain hardening, and stabilizes the plastic flow. It should be emphasized that, despite the relative softening, the nanoparticles' strength remains on the level of tens of GPa. This combination of ultra-high strength and increased toughness can make the alloy nanoparticles promising candidates for many technological, biological, and medical applications[27–29], including nanoelectronics[30], nanomagnetics[31,32], and catalysis[33,34]. In particular, the catalytic activity of metallic nanoparticles can be optimized by strain engineering[35–37]. For example, the catalytic activity can be enhanced by coherency strains arising in the shell of core-shell particles[36], but only if the material can sustain high enough stresses. The synergism between the strength and toughness achieved by alloying can open new opportunities for the design of metallic nanoparticles with advanced catalytic performance.

The failure of both Ni and Ni-Co nanoparticles is caused by a dislocation avalanche triggered by a single dislocation nucleation event at one of the facets. However, the detailed dislocation mechanisms operating in the Ni and Ni-Co particles are

markedly different. In Ni particles, the first nucleation event is immediately followed by the nucleation of multiple dislocations from the same location, creating a dislocation crowd quickly spreading into the particle. In the alloy particles, the first dislocation evolves into a complex shape consisting of curls and spirals forming a thick tangle propagating deep into the particle's interior. Pinning by solute atoms breaks the dislocation line into multiple irregular segments and impedes its motion (Supplementary Fig. 4).

An interesting feature of the alloying effect found here is the non-monotonic dependence of the particle strength on the solute concentration. Simulations show that the particle strength drops about 33% relative to pure Ni upon the addition of about (0.10–0.15) Co atoms (Fig. 5c) before rebounding at higher concentrations. We have tested the hypothesis that this effect is caused by elastic softening of the material evident from the slopes of the initial portions of the stress-strain plots (Fig. 5a). Calculations show that the effective modulus of the particles does indeed follow a non-monotonic behavior with chemical compositions, as does the Young modulus of the bulk Ni and Ni-Co in the [111] direction (Supplementary Fig. 5). However, the elastic minimum occurs at about (0.20–0.25) Co and is too shallow (about 12% relative to pure Ni) to explain the deep minimum of the strength. A more plausible explanation is that the minimum of the strength is caused by competition between two opposing factors: the solute-induced reduction of the nucleation barrier for the first dislocation and the solute friction impeding the dislocation motion once the nucleation barrier is overcome. A minimum can arise if the first effect is stronger at lower Co concentrations (causing softening) while the second dominates at higher concentrations (causing hardening).

The solute softening can be explained by spatial variations of the local stress caused by the random distribution of the solute atoms. To demonstrate such variations, we examined the distribution of the maximum resolved atomic stress (MRAS) across the particles right before the nucleation of the first dislocation (see the "Methods" section). We found that, in the alloy particles, the distribution exhibits a long tail extending to high MRAS values (Fig. 8a). (For visual clarity, the plot in Fig. 8a uses MRAS

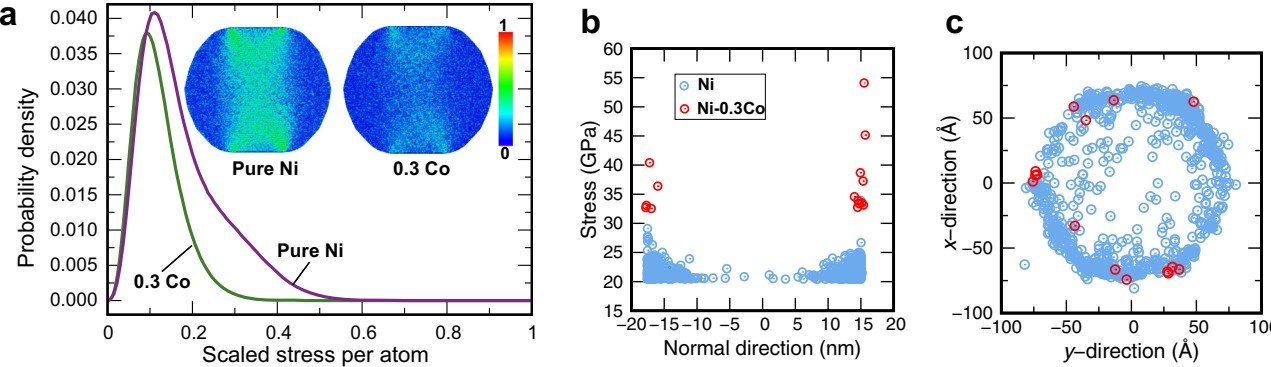

**Fig. 8 Characterization of stresses in nanoparticles under compression. a** MRAS distribution scaled by the min-max formula (1) in 35 nm particles of pure Ni and Ni-0.3Co. The inset shows the min−max scaled MRAS distributions in the particle cross-sections. **b** MRAS in the particles as a function of the $z$-coordinate normal to the substrate ($z = 0$ at the particle center). Only the high end of the MRAS distribution is shown for clarity. **c** Positions of the points shown in (**b**) in the top view of the particle. The hexagon shape corresponds to the top (111) facet. Note that the highest MRAS values are reached in the Ni-0.3Co particle and are concentrated at the facet edges (red circles).

values scaled by the maximum value as explained in the "Methods" section. The high-stress tail existing in the alloy particles pushes the peak of the distribution curve toward smaller scaled MRAS values.) The high MRAS values are predominantly reached near the edges of the top or bottom facet (Fig. 8b, c). As a result, the critical stress for the dislocation nucleation is reached when the average stress in the particle is lower than in Ni particles of the same size. A similar softening model was discussed in recent simulations of Fe-Ni nanowires[17], where local pressure was used instead of the more relevant MRAS.

On the hardening branch of the strength curve (Fig. 5c), we observe solute pinning of the dislocation lines suggesting significant friction to their motion. The dislocations also undergo more extensive cross-slip in the alloy particles than in Ni particles (Supplementary Fig. 6). The cross-slip is activated by the increased stacking fault energy (SFE) in the random Ni-Co solution compared to pure Ni. This is evident from the narrower dissociation widths in the alloy particles (Fig. 6) and was also confirmed by direct calculations. Supplementary Fig. 7 shows that the SFE significantly increases with Co concentration, reaching a shallow maximum at about (0.4–0.5) Co. There is no contradiction to the experimentally observed SFE reduction in Ni-Co alloys[2]. The measurements were made on annealed samples with an equilibrium distribution of the solute atoms at the stacking faults (Suzuki atmospheres). In our case, the fast-moving dislocations do not have the time to reduce the SFE by forming Suzuki atmospheres, which is a diffusion-controlled process. Combined with the solute pinning effect, the extensive cross-slip leads to highly entangled dislocation configurations with relatively low mobility.

## Methods

**Experimental methodology.** Ni-Co bilayers with the Ni layer on top were deposited on the (0001)-oriented sapphire substrate by magnetron sputtering under Ar (99.9999%) atmosphere of 0.4 Pa and room temperature without vacuum break (Fig. 1a). The polished side of the substrate had been ultrasonically cleaned prior to the deposition following the standard cleanroom protocol. The initial film thickness influences the size distribution of the nanoparticles obtained after the dewetting. To produce particles with consistent size distributions, the film thickness was kept constant at 30 nm for different chemical compositions, including pure Ni. The films were heat-treated under forming gas atmosphere (Ar-10% $H_2$, flow rate 100 sccm) in a resistive tube furnace GSL-1500X-OTF (MTI corporation). A bare sapphire plate was placed between the sample and the quartz boat to avoid any source of contamination. The dewetting was performed at the temperature of 1150 °C for 18 h (Fig. 1b). A low heating and cooling rate of 4 °C/min was chosen to avoid thermal shock and achieve complete mixing in the bilayer.

The morphology of the nanoparticles produced by the dewetting was characterized using Carl Zeiss Ultra Plus high-resolution scanning electron

microscope (HR-SEM) at an operating voltage of 3 keV. The cross-sectional microstructure of the Ni-0.3Co and Ni-0.5Co particles was further characterized by transmission electron microscopy (TEM). Bright-field TEM micrographs and selected-area electron diffraction (SAED) patterns were acquired in FEI Technai G2 T20 TEM operating at 200 keV. Elemental distribution maps were recorded using energy dispersive X-ray spectroscopy (EDS) in the scanning transmission electron microscopy (STEM) mode in aberration-corrected Themis G2 300 TEM operating at 300 keV. All cross-sectional TEM samples were prepared using the standard lift-out method in FEI Helios Nanolab G3 Dualbeam FIB. The lift-out lamella was mounted on Mo grid and thinned down by Ga ion beam to electron transparency. The lamella was finally cleaned by Ga ion beam with a current of 7 pA at 2 keV to remove damaged layers.

In-situ compression tests of the nanoparticles were performed using a Hysitron PI85 picoindenter with a flat rectangular diamond punch of 1 μm in diagonal actuated by capacitance-controlled transducer inside a HR-SEM. The tests were conducted in displacement-control mode at a constant displacement rate of 1 nm/s. Load-displacement data from the compression tests were obtained after correcting for thermal drift. The image analysis was performed using ImageJ20 software. TEM and STEM EDS data were visualized and analyzed using Gatan Microscopy Suite (GMS3) DigitalMicrograph® and Thermo Scientific Velox®, respectively.

**Simulation methodology.** Atomic interactions in the Ni-Co system were modeled by the many-body angular-dependent interatomic potential[26]. The MD simulations utilized the Large-scale Atomic/Molecular Massively Parallel Simulator (LAMMPS)[38]. The atomic structures were visualized with the Open Visualization Tool (OVITO) software package[39]. The initial set of Ni nanoparticles was created by the Wulff construction using the 0 K surface energies predicted by the potential. The particle sizes ranged from 15 to 60 nm ($10^5$–$10^7$ atoms). The initially sharp edges and corners of the particles were smoothed by the "simulated surface evaporation" procedure developed previously[9]. We chose the roundness parameter[9] of 0.03 to best represent the experimental particle shapes. To obtain the alloy particles, we randomly replaced the Ni atoms with Co to achieve the desired chemical composition.

In the simulated compression tests, the substrate and indenter were represented by a harmonic potential wall exerting a linear force with an effective elastic modulus of 100 GPa. The position of the lower wall (substrate) was fixed while the upper wall (indenter) was moved into the particle with a constant speed of 1 m s$^{-1}$, simulating the displacement-controlled deformation implemented in the experiments. The top and bottom facets of the particle had the (111) orientation. To eliminate thermal stresses, we pre-expanded the particles according to the composition-dependent thermal expansion factor at the MD temperature (300 K) and thermalized them by a canonical MD simulation before applying the load. The engineering stress and strain were defined as in the experiment. The stress distribution inside the particle was represented by the virial stress tensor implemented in LAMMPS. As in the experiments, we define the particle strength as the engineering stress at the first peak of the stress-strain curve, and the particle size as its initial effective diameter. The top facet area was defined by $nA_{(111)}$, where $n$ is the number of atoms on the facet, $A_{(111)} = a^2\sqrt{3}/4$ is the area per atom in the (111) crystallographic plane of the FCC structure, and $a$ is the cubic lattice parameter. The strength values reported here were obtained by averaging over several (5–10) independent tests with different random distributions of the Co atoms. The error bars in Fig. 5b,c correspond to one standard deviation.

The atomic structures were visualized by several different methods, depending on the goal. These include visualization by the potential energy, the bond-angle

analysis, and the dislocation extraction algorithm (DXA) implemented in OVITO. The DXA algorithm was used to visualize and count the dislocations inside the particles (Fig. 6c) and analyze the dislocation cross-slip (Supplementary Fig. 4c).

The effective elastic modulus of the nanoparticles reported in Supplementary Fig. 5 was computed from the slopes of the initial portions of the stress-strain curves (see for example Fig. 5a). The Young modulus $Y_{[111]}$ of bulk Ni and Ni-Co alloys was obtained by MD simulations of a periodic cell (about $7 \times 10^5$ atoms) elongated along the [111] direction with a strain rate of $10^{-7}$ s$^{-1}$ under zero stress conditions in the lateral directions. $Y_{[111]}$ was extracted from linear fits to the stress-strain relations averaged over several random distributions of the Co atoms.

To study the stacking faults, four parallel edge dislocations were created in a square periodic block (about $10^5$ atoms), as in Supplementary Fig. 7a,b. After introducing Co atoms by random substitution, and 4 ns long MD run was performed at 300 K to allow equilibrium dissociation of the dislocations into partials. The stacking fault width was estimated from the number of atoms with a hexagonal close-packed environment identified by the bond angle analysis. The stacking fault energy was obtained by separate calculations using a simulation block (about $2 \times 10^5$ atoms) containing a single stacking fault with open surfaces in the direction normal to the fault. The excess energy associated with the fault was calculated by comparing the total energy with the energy of a perfect lattice block containing the same number of atoms and the same open surfaces. The results reported in Supplementary Fig. 7c,d were obtained by averaging over several Co atom distributions.

The stress distribution inside the particles was analyzed in terms of the maximum resolved atomic stress (MRAS), defined as the maximum value of the atomic stress tensor resolved along all possible slip systems. The min–max scaled MRAS (Fig. 8) was obtained from the maximum and minimum MRAS values in the particle by the formula

$$(\text{MRAS})_{\text{scaled}} = \frac{\text{MRAS} - (\text{MRAS})_{\text{min}}}{(\text{MRAS})_{\text{max}} - (\text{MRAS})_{\text{min}}}. \tag{1}$$

The results reported in Fig. 8a were averaged over 5 independent simulations with different distributions of Co atoms. The lower threshold MRAS values used in Fig. 8b, c are 20.4 GPa for Ni and 32.5 GPa for Ni-0.3Co.

Since MD simulations were heavily involved in this work, comments are in order about the well-known length- and timescale limitations of the method. The particle sizes used in the MD simulations were smaller than in the experiment. However, this allowed us to extend the work toward smaller particles whose strength could not be readily measured by experiment. The simulations have also provided detailed information about the dislocation nucleation mechanisms that lie beyond the experimental capabilities. There is also a significant gap between the simulated and experimental strain rates. This gap would be crucial if we studied diffusion-controlled processes, such as diffusional creep, dislocation climb, or solute drag. However, in this work, we studied processes occurring on shorter timescales. Until the particle reaches the critical stress, the deformation is elastic and thus readily reproducible by MD. The dislocations nucleate when the barrier is suppressed to nearly zero, not requiring a significant thermal activation. The solute softening effect is explained by static variations in the local stress due to the randomness of the solute atom positions. The timescale is not a critical factor in this mechanism. Once the dislocation has nucleated, it zooms through the particle with speed close to the speed of sound. Diffusion or any other thermally activated processes do not have the time to occur. Our MD simulations readily capture this mode of deformation.

## Data availability

The data that support the findings of this study are available in the Supplementary Information file or from the corresponding authors upon reasonable request. The computer simulation part of this research used the publicly available codes LAMMPS and OVITO. The routine computer scripts controlling the execution of the calculations are not central to this work but are available from the corresponding authors upon reasonable request.

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

## Acknowledgements

R.K.K. and Y.M. acknowledge support from the National Science Foundation, Award No. 1904428. E.R., A.B., and Y.Q. wish to thank the US-Israel Bi-National Science Foundation (joint BSF-NSF grant No. 2018625) for support. J.H. was supported by an NRC Research Associateship award at the National Institute of Standards and Technology (NIST). The commercial names are used in this paper for completeness only and do not constitute an endorsement from NIST.

## Author contributions

A.B. conducted the experiments designed and directed by E.R.Y.Q. contributed to TEM characterization of the particles. R.K.K. conducted the simulations under Y.M.'s direction and supervision. J.H. help R.K.K. set up the simulation methodology at the initial stages of this work. Y.M. prepared a draft of the complete paper. All co-authors participated in the paper editing and approved its final version.

## Competing interests

The authors declare no competing interests.
