## [Peer Review File · Nature Communications]

REVIEWER COMMENTS

Reviewer #1 (Remarks to the Author):

In their manuscript “Softer but tougher: The impact of alloying on defect-free nanoparticles”, the authors present the surprising effect that alloying has on nanoparticles mechanical response, in contradiction with what is known for bulk materials. By direct experimental observations and measurements, supported and enlighten by atomistic simulations, the authors demonstrate that alloying Ni particles with Co has a strong non-monotonous influence on their strength. On the opposite, alloyed particle requires more work to further deform plastically, that is directly related to an increase in toughness. This behavior is presented as being the result of two competitive mechanisms: stress concentration that favor dislocation nucleation at “lower” stress, and solute friction that limit dislocation motion within the particle. I found particularly elegant the way the authors demonstrate the “local stress concentration” induced by the alloying elements: by using the “probability density” of the “maximum resolved atomic stress”.

The manuscript is very well written and structured. It is very pleasant to read, especially the clear and comprehensive introduction. By considering mostly the simulation part, most of the questions that arise during the reading are answered at a later stage, or by looking at the method section. I particularly appreciated the great care taken in the simulations setup (i.e. thermal equilibration), or to ensure the statistical relevance of the numerical results.

I have one concern regarding the manuscript. The discussion says, “in the alloy particles, the distribution exhibits a long tail extending to high MRAS values”. This is clear and fitting with all the demonstration. However, Fig. 8(a) seems to show the exact opposite. I assumed that it was a mistake in the figure while reading the manuscript. The authors should either correct this very important figure or provide more explanation. Similarly, in Fig. 8(b,c), it seems that there is much more data point for the pure Ni particle than for the alloyed one. A brief comment in the caption itself will clarify this aspect.

Reviewer #2 (Remarks to the Author):

The authors discuss the impact of alloying on the mechanical properties of nickel-based nanoparticles. The binary system the authors decided to study is Ni-Co. The solvent is played by Ni while the solute element is played by Co. The solute effect on the strength of defect-free metallic nanoparticles has never been studied experimentally or by simulation. The paper contains 39 references and 8 figures. The bibliography is rich and up to date. Therefore, this paper is highly desirable to the community.

Here are some questions/remarks:

1. Why did you choose the Ni-Co system?
2. Do you expect to get the same conclusions with a purely bcc binary alloy?
3. On page 7, the authors claim that "The experiments and simulations reported here demonstrate that

alloying of defect-free Ni nanoparticles with Co reduces their strength. This counter-intuitive solid-solution softening effect contradicts the established behavior of bulk alloys, which are dominated by the solid-solution hardening [20]." I think the reduction in strength is directly correlated to the particular binary phase diagram of Ni-Co at the nanoscale (Ref. 32). Nickel has a higher melting point than Cobalt at small sizes, meaning that nickel has a larger cohesive energy than cobalt, therefore there is nothing surprising or counter intuitive that by adding cobalt to a nickel solvent, you are reducing its strength. This can be explained by looking at the binary phase diagram of Ni-Co at the nanoscale. Of course, if you are considering the bulk phase diagram of Ni-Co; then, I agree it is counter-intuitive. Maybe, you could emphasize that point.

Reviewer #3 (Remarks to the Author):

The manuscript deals with the surprising mechanical properties of nanometer scale metallic particles and especially the effect of alloying Ni with Co as compared to pure Ni nanoparticles. Whereas in the bulk materials, alloying increases the strength, the present study shows that it is different in nanoparticles. If we recall that the strength of defect-free faceted nanoparticles exhibits an ultra-high strength approaching the theoretical limit, we can ask if alloying will go over the limit or rather go in the opposite direction. The authors show by experimental and theoretical studies that indeed alloying reduces the ultimate strength, which is counter-intuitive with the solution-hardening observed in the bulk alloys and nanowires. The authors explain why different mechanisms in bulk and in defect-free nanoparticles leads to different behaviours. Moreover they show that the alloy nanoparticles become tougher, which makes a good compromise for interesting applications. In view of the large panel of industrial applications it is very important to characterize their mechanical properties to know under which mechanical solicitations they can resist.

In that sense, this study brings very interesting informations by coupling experimental and numerical simulations to characterize and understand the atomistic mechanisms responsible of the macroscopic behaviour, which is that before a certain charge in compression, the nanoparticles are elastically deformed, then, they collapse into a pancake shape. MD simulations reveal in particular that the softening in defect-free nanoparticles is due to statistical variation of the local resolved shear stress near the surface, inducing early nucleation of the first dislocation.

The manuscript is very clear and well illustrated by the figures. I would only have some remarks that the authors could comment in order to better clarify some points but the overall results presented in this study is of significant relevance and novelty to be published in Nature Communications.

Here are my remarks:

1. The discrepancy in the size between experimental samples and theoretical systems (the size of the nanoparticles in MD simulations is one order of magnitude smaller than in the experiments) are questionable even if we know the limit of the numerical calculations but the authors could comment on

it.

2. in the same idea, the gap between the constant speed of displacement of the indenters in experiments (1 nm/s) and in the simulations (1 m/s) should be commented even if I am aware of that it is not possible to decrease the speed in simulations for computational reasons.

3. The plastic deformation in the pancake shape in Fig. 7 for pure Ni as compared to Ni-Co nanoparticles suggest a quite important effect of Co alloying in the toughness and also in the homogeneity of the final shape. However I wonder if the experiments could report the same effect ? And what is the influence of the interatomic potential, comparing pure Ni and Ni-Co potentials because for such important deformations, the two kinds of potentials should have been tested in situations where the atomic positions are far from their equilibrium position. In other words, for example the cut-off distances for the interactions could have a non negligible impact on the results so it would have been relevant to use potentials which are well comparable (which is perhaps the case but not mentioned).

Point-by-point response to Reviewer's comments

“Softer but tougher: The impact of alloying on defect-free nanoparticles”

By Anuj Bisht, Raj Kiran Koju, Yuanshen Qi, James Hickman, Yuri Mishin and Eugen Rabkin.

Manuscript NCOMMS-21-00133

February 8, 2021

We are grateful to the Reviewers for carefully reading the manuscript and providing insightful comments. The paper has been revised to address the suggestions made by the Reviewers. This document summarizes our responses and changes made in the manuscript, which are highlighted in red.

Reviewer #1 (Remarks to the Author):

In their manuscript “Softer but tougher: The impact of alloying on defect-free nanoparticles”, the authors present the surprising effect that alloying has on nanoparticles mechanical response, in contradiction with what is known for bulk materials. By direct experimental observations and measurements, supported and enlighten by atomistic simulations, the authors demonstrate that alloying Ni particles with Co has a strong non-monotonous influence on their strength. On the opposite, alloyed particle requires more work to further deform plastically, that is directly related to an increase in toughness. This behavior is presented as being the result of two competitive mechanisms: stress concentration that favor dislocation nucleation at “lower” stress, and solute friction that limit dislocation motion within the particle. I found particularly elegant the way the authors demonstrate the “local stress concentration” induced by the alloying elements: by using the “probability density” of the “maximum resolved atomic stress”.

The manuscript is very well written and structured. It is very pleasant to read, especially the clear and comprehensive introduction. By considering mostly the simulation part, most of the questions that arise during the reading are answered at a later stage, or by looking at the method section. I particularly appreciated the great care taken in the simulations setup (i.e. thermal equilibration), or to ensure the statistical relevance of the numerical results.

Comment1: I have one concern regarding the manuscript. The discussion says, “in the alloy particles, the distribution exhibits a long tail extending to high MRAS values”. This is clear and fitting with all the demonstration. However, Fig. 8(a) seems to show the exact opposite. I assumed that it was a mistake in the figure while reading the manuscript. The authors should either correct this very important figure or provide more explanation. Similarly, in Fig. 8(b,c), it seems that there is much more data point for the pure Ni particle than for the alloyed one. A brief comment in the caption itself will clarify this aspect.

Response1: The distributions shown in Fig.8(a) are labeled correctly. As indicated in the figure caption, the stress plotted on the x-axis is the stress **scaled** by its maximum value. Since the maximum stress in the alloy particles is significantly higher, the respective normalized distribution is shifted toward smaller normalized values, revealing that the shape of the curve is dominated by a long tail. We have added an explanation of this graphical form in the second paragraph on page 8.

Regarding Figures 8(b,c), for visual clarity, we only show the points for stresses above a threshold value. These values are indicated in the Methods section and were adjusted to best reveal the effects we wanted to demonstrate. This explains the different numbers of points mentioned by the Reviewer. We believe that this is clearly explained in the paper.

Reviewer #2 (Remarks to the Author):

The authors discuss the impact of alloying on the mechanical properties of nickel-based nanoparticles. The binary system the authors decided to study is Ni-Co. The solvent is played by Ni while the solute element is played by Co. The solute effect on the strength of defect-free metallic nanoparticles has never been studied experimentally or by simulation. The paper contains 39 references and 8 figures. The bibliography is rich and up to date. Therefore, this paper is highly desirable to the community.

Here are some questions/remarks:

Comment1: Why did you choose the Ni-Co system?

Response1: As mentioned in the Introduction section of the manuscript, Co exhibits unlimited solubility in the face-centered cubic (FCC) Ni. This allows us to focus this research on the solid-solution effect uninfluenced by the precipitation hardening and other mechanisms of alloy strengthening. The literature sources cited indicate that the bulk Ni(Co) alloys follow the classical pattern of solid solution hardening. Also, previous computational results (see Ref. [21] in the manuscript) indicate that similar behavior can be expected in defect-free alloy nanowires. Since the mechanical properties of pure Ni nanoparticles have been thoroughly investigated in our previous work (Ref. [9] in the manuscript), studying the nanoparticles of a Ni-based alloy was a natural choice. In the cited work [9], it was shown that the Ni nanoparticles were stronger than all other sub-micrometer sized FCC metal objects studied so far. Thus, our initial hope was that the Ni(Co) alloy particles would exhibit even higher strength, setting a new record of metals strength. Our experimental results proved the opposite, and the atomistic simulations enabled us to understand the mechanisms of the solid solution softening effect.

A sentence explaining our choice of this system has been added in the second paragraph on p.3.

Comment2: Do you expect to get the same conclusions with a purely bcc binary alloy?

Response2: Our recent results on compression of nano- and microparticles of BCC Mo (see Ref. [10] in the manuscript) indicate that similar to FCC metals, their deformation mechanisms are governed by the laws of dislocation nucleation-controlled plasticity. Based on this similarity, we expect a similar solid solution softening effect in BCC alloys. Nonetheless, considering the significant differences in the structure of dislocation cores in the FCC and BCC metals and the resulting dislocation slip mechanisms, some surprises may be discovered in BCC alloy nanoparticles.

Comment3: On page 7, the authors claim that "The experiments and simulations reported here demonstrate that alloying of defect-free Ni nanoparticles with Co reduces their strength. This counter-intuitive solid-solution softening effect contradicts the established behavior of bulk alloys, which are dominated by the solid-solution hardening [20]." I think the reduction in strength is directly correlated to the particular binary phase diagram of Ni-Co at the nanoscale (Ref. 32). Nickel has a higher melting point than Cobalt at small sizes, meaning that nickel has a larger cohesive energy than cobalt, therefore there is nothing surprising or counter intuitive that by adding cobalt to a nickel solvent, you

are reducing its strength. This can be explained by looking at the binary phase diagram of Ni-Co at the nanoscale. Of course, if you are considering the bulk phase diagram of Ni-Co; then, I agree it is counter-intuitive. Maybe, you could emphasize that point.

Response3: Indeed, according to Ref. [32] of the manuscript, the depression of the melting point of Co nanoparticles should be much stronger than for Ni nanoparticles. This effect is predicted based on the anomalously high difference between the surface energies of solid and liquid Co. However, the solid surface energy used in this prediction [Vitos et al., Surface Sci. 411 (1998) 186] is unreliable. In any case, the melting point depression in nanoparticles is of little relevance in our work because the lateral size of the experimental particles studied here was larger than 100 nm. According to Ref. [32], the melting point depression of particles with such dimensions is negligible in both Ni and Co.

We would also like to emphasize that in the dilute binary M(X) systems (where M is the matrix and X is the solute), the strength of pure X is not directly related to the strengthening effect of the X atoms as a solute in the M matrix. In this respect, the Cu-Sn system is a classic example of solute strengthening that gave its name to an important period of human history (the bronze age). The Cu-Sn bronze is much stronger than pure Cu, even though the melting point of Sn (232 °C) is significantly lower than that of Cu (1085 °C).

Reviewer #3 (Remarks to the Author):

The manuscript deals with the surprising mechanical properties of nanometre scale metallic particles and especially the effect of alloying Ni with Co as compared to pure Ni nanoparticles. Whereas in the bulk materials, alloying increases the strength, the present study shows that it is different in nanoparticles. If we recall that the strength of defect-free faceted nanoparticles exhibits an ultra-high strength approaching the theoretical limit, we can ask if alloying will go over the limit or rather go in the opposite direction. The authors show by experimental and theoretical studies that indeed alloying reduces the ultimate strength, which is counter-intuitive with the solution-hardening observed in the bulk alloys and nanowires. The authors explain why different mechanisms in bulk and in defect-free nanoparticles leads to different behaviours. Moreover they show that the alloy nanoparticles become tougher, which makes a good compromise for interesting applications. In view of the large panel of industrial applications it is very important to characterize their mechanical properties to know under which mechanical solicitations they can resist.

In that sense, this study brings very interesting information's by coupling experimental and numerical simulations to characterize and understand the atomistic mechanisms responsible of the macroscopic behaviour, which is that before a certain charge in compression, the nanoparticles are elastically deformed, then, they collapse into a pancake shape. MD simulations reveal in particular that the softening in defect-free nanoparticles is due to statistical variation of the local resolved shear stress near the surface, inducing early nucleation of the first dislocation.

The manuscript is very clear and well illustrated by the figures. I would only have some remarks that the authors could comment in order to better clarify some points but the overall results presented in this study is of significant relevance and novelty to be published in Nature Communications.

Here are my remarks:

Comment1: The discrepancy in the size between experimental samples and theoretical systems (the

size of the nanoparticles in MD simulations is one order of magnitude smaller than in the experiments) are questionable even if we know the limit of the numerical calculations but the authors could comment on it.

Response1: We are aware that the particles used in the MD simulations were smaller than the experiment due to the computational limitations of MD. This is state of the art in the MD simulation field that we cannot change. On the other hand, the simulations extend this work toward smaller particles whose strength could not be easily measured by experiments. The simulations also provide detailed information about the dislocation nucleation mechanisms that lie beyond the experimental capabilities. Thus, the synergism of the experimental study and the MD modeling enabled us to reach the level of understanding unattainable by each method alone.

Comment2: in the same idea, the gap between the constant speed of displacement of the indenters in experiments (1 nm/s) and in the simulations (1 m/s) should be commented even if I am aware of that it is not possible to decrease the speed in simulations for computational reasons.

Response2: Like in the previous comment, we agree that a gap exists between the simulated and experimental strain rates. This gap would indeed be crucial if we studied diffusion-controlled processes, such as diffusional creep, dislocation climb, or solute drag. These processes occur on timescales much longer than the timescales accessible by MD simulations. However, in this work, the simulation results are highly relevant to the experimental conditions. Until the particle reaches the critical stress, the deformation is elastic and thus readily reproducible by MD. The dislocation nucleation occurs when the barrier is suppressed to nearly zero, not requiring a significant thermal activation. Furthermore, the explain the solute softening effect by static variations of the local stress arising due to the randomness in the solute atom positions. The timescale is not a critical factor in this mechanism. Finally, once the dislocation nucleation has occurred, the dislocations zoom through the particle extremely fast, probably with speeds close to the speed of sound. Diffusion or any thermally activated processes do not have the time to occur. This mode of deformation is readily captured by MD simulations. The abrupt collapse of the particles observed in our experiments confirms the athermal nature of their deformation. We would also like to draw the Reviewer's attention to the recent work [A. Sharma et al, "Pseudoelasticity of metal nanoparticles is caused by their ultra-high strength", *Advanced Functional Materials* **30** (2020) 1807554] in which the conditions at which the surface and interface diffusion may play a significant role in the particle deformation were established. The size of the Ni and Ni-Co particles studied in the present work (above 100 nm) safely excludes any role of long-range surface/interface diffusion in the deformation process.

We have followed the Reviewer's suggestion and added these comments in the Methods section.

Comment3: The plastic deformation in the pancake shape in Fig. 7 for pure Ni as compared to Ni-Co nanoparticles suggest a quite important effect of Co alloying in the toughness and also in the homogeneity of the final shape. However I wonder if the experiments could report the same effect ? And what is the influence of the interatomic potential, comparing pure Ni and Ni-Co potentials because for such important deformations, the two kinds of potentials should have been tested in situations where the atomic positions are far from their equilibrium position. In other words, for example the cut-off distances for the interactions could have a non negligible impact on the results so it would have been relevant to use potentials which are well comparable (which is perhaps the case but not mentioned).

Response3: The compression experiments were performed using Hysitron PI85 *in-situ* picoindenter equipped with a diamond punch actuated by a capacitance-controlled transducer. The natural working mode of this transducer is load-control, since electrical signal on the transducer's capacitor is directly translated into the loading force. The displacement-control mode is realized via an electronic feedback loop system. All our experiments were performed in displacement-control mode, in which the punch is lowered and retracted with a constant preset rate (typically, 1 nm/s). At the onset of yielding, massive nucleation of dislocations occurs in the particle, resulting in a sudden loss of its strength. These events occur on a timescale that is much shorter than the time constant of the feedback loop system. As a result, the punch smashes the particle into a disc shape with a high force close to the maximum loading force at the onset of the yielding. This prevents the experimental system from measuring the actual flow stress as a function of displacement, and thus the toughness. Nonetheless, we noted that, beyond the yield point, smaller strain bursts (on average) are observed in Ni-Co alloy particles than in pure Ni particles (see Fig. 3a-c in the manuscript). This behavior is consistent with the results of our atomistic simulations in two ways: firstly, the simulations indicate that fewer dislocations are nucleated in the Ni-Co particles than in the Ni particles; secondly, the simulations demonstrate that Co atoms impede the dislocation propagation in the Ni-Co particles, making it more likely that the particle will regain the ability to resist the compression. Because of the difference in the amplitude of displacement bursts, the compressed Ni particles are flatter and exhibit a smaller final thickness (Fig. 1a below) compared to the compressed Ni-50Co particles (Fig. 1b below). This indirectly confirms the higher degree of strain-hardening in Ni-Co alloy particles. Material with higher strain hardening is expected to show larger toughness under true displacement-control compression. Nonetheless, the compressed particles of both Ni and Ni-50Co alloy appear to be homogeneously deformed (see Fig. 1 below). Due to the uncontrollable movement of the punch upon yielding, subtle differences in the morphology of compressed Ni and Ni-Co particles at the initial stages of plasticity cannot be captured experimentally.

Fig. 1 Representative SEM micrographs of the particles after compression: (a) Ni, and (b) Ni-50Co.

Regarding the interatomic potential question, the Ni, Co, and Ni-Co potentials have been extensively tested for high isotropic and tensile deformations during their developments. Other groups have successfully used the Ni-Co potential in simulations of mechanical behavior of Ni-Co alloys, see for example:

K.V. Reddy and S. Pal: “Analysis of deformation behaviour of Al–Ni–Co thin film coated aluminium during nano-indentation: a molecular dynamics study”, *Molecular Simulation* **44**, 1393 (2018).

Yuan et al.: “Nano-twinning in a γ' precipitate strengthened Ni-based superalloy”, *Materials Research Letters* **6**, 683 (2018).

Xu et al.: “Tensile mechanical performance of Ni–Co alloy nanowires by molecular dynamics simulation”, *RCS Advances* **9**, 25817-25828 (2019).

Ali Imram et al.: “Deformation Mechanisms and Dislocations in Nickel–Cobalt Core–Shell Nanowires Under Uniaxial Tensile Loading—A Molecular Dynamics Modeling Analysis”, *Advanced Science, Engineering and Medicine* **11**, 1187 (2019).

We are confident that the interatomic potential used in this work is accurate enough to provide reliable results for deformation behavior of Ni-Co alloys.

Sincerely,

Anuj Bisht, Raj Kiran Koju, Yuanshen Qi, James Hickman, Yuri Mishin and Eugen Rabkin.

REVIEWERS' COMMENTS

Reviewer #3 (Remarks to the Author):

The authors answered very precisely to the questions and comment so in my opinion the manuscript is now suitable for publication in Nature Communication.